

# Development rate rather than social environment influences cognitive performance in Australian black field crickets, *Teleogryllus commodus*

Caitlin L. Anderson and  Michael M. Kasumovic

Ecology & Evolution Research Centre, UNSW Australia, Sydney, NSW, Australia

## ABSTRACT

Cognitive functioning is vital for enabling animals of all taxa to optimise their chances of survival and reproductive success. Learning and memory in particular are drivers of many evolutionary processes. In this study, we examine how developmental plasticity can affect cognitive ability by exploring the role the early social environment has on problem solving ability and learning of female black field crickets, *Teleogryllus commodus*. We used two learning paradigms, an analog of the Morris water maze and a novel linear maze, to examine cognitive differences between individuals reared in two acoustic treatments: silence or calling. Although there was no evidence of learning or memory, individuals that took longer to mature solved the Morris water maze more quickly. Our results suggest that increased investment into cognitive development is likely associated with increased development time during immature stages. Inconsistent individual performance and motivation during the novel linear maze task highlights the difficulties of designing ecologically relevant learning tasks within a lab setting. The role of experimental design in understanding cognitive ability and learning in more natural circumstances is discussed.

## INTRODUCTION

Vital tasks for all animals, such as survival and reproduction, are facilitated by their ability to perceive and interpret their world and are therefore heavily reliant on cognitive functioning (*Buchanan, Grindstaff & Pravosudov, 2013*). Learning and memory are particular drivers of many evolutionary processes as they allow individuals to effectively respond to cues within varied environments and respond accordingly (see *Mettke-Hofman, 2014* for review), potentially through the use of novel behaviours (*Dukas, 2013*). For example, individuals demonstrate environmentally-cued condition-dependent cognitive development (see *Buchanan, Grindstaff & Pravosudov, 2013* for review) when novel behaviours may be useful to navigate more complex environments (*Clayton & Krebs, 1994*) that can vary due to seasonality (*Clayton, Reboreda & Kacelnik, 1997*), food availability (*Healy & Krebs, 1992*), and urbanisation (*Mettke-Hofmann, 2016*). However, due to the large energy investment needed to develop and maintain the cognitive structures needed for learning, studies

Corresponding author
Michael M. Kasumovic,
m.kasumovic@unsw.edu.au

demonstrate that investment into learning often reduces fitness related traits such as reproductive success (*Snell-Rood, Davidowitz & Papaj, 2011*), adult (*Burger et al., 2008*) and larval longevity (*Mery & Kawecki, 2003*), locomotor activity (*Zwoinska et al., 2016*) or fecundity (*Mery & Kawecki, 2004*).

Due to the impact this plasticity has on an individual's fitness and therefore the capability of a population or species to withstand rapid environmental shifts (*Healy & Braithwaite, 2000*; *Roth, Krochmal & Németh, 2015*), it may only be beneficial to invest in cognitive capacity during development when there is a potential fitness benefit (*Buchanan, Grindstaff & Pravosudov, 2013*; *Van Praag, Kempermann & Gage, 2000*). It is thus important to understand the various environmental factors that can affect how immature individuals allocate resources during development and how changes in allocation strategies shape cognitive performance at maturity (*Buchanan, Grindstaff & Pravosudov, 2013*). Along those lines, there are many studies that demonstrate how environmental variation in food types (*Kotrschal & Taborsky, 2010*), habitat complexity (*Sheenaja & Thomas, 2011*), or visual environments (*Girvan & Braithwaite, 2000*), during the juvenile stage can trigger differences in how resources are allocated towards cognitive development and other fitness related traits.

Developmental plasticity, however, isn't limited to environmental factors and an equally important aspect in determining how individuals allocate resources towards various traits is the adult social environment (*Kasumovic & Brooks, 2011*). Although research is uncovering how social environments alter developmental trajectories (*DiRienzo, Pruitt & Hedrick, 2012*; *Kasumovic et al., 2011*; *Kasumovic & Andrade, 2006*; *Kasumovic & Andrade, 2009*), adult behaviour (*Bailey, Gray & Zuk, 2010*; *Kasumovic, Hall & Brooks, 2012*), and personality (*DiRienzo, Pruitt & Hedrick, 2012*), the explorations have been limited to quantitative and behavioural traits. The primary objective of this study is thus to explore whether the social environment also influences cognitive development in a similar manner to other environmental cues. To explore this question, we use the Australian black field crickets (*Teleogryllus commodus*).

The social environment is of particular importance to *T. commodus* as individuals alter their investment in fitness related traits and behaviours in response to the density and quality of calls they hear while in their final juvenile instar (*Kasumovic et al., 2011*). The density and quality of calls heard prior to maturity affects how males call to attract females at maturity and also affects how females vary in their motivation to choose a mate (*Kasumovic et al., 2011*; *Kasumovic, Hall & Brooks, 2012*). The latter is of particular interest with respect to cognitive development as female fitness relies on the ability to successfully locate a calling male (*Loher & Rence, 1978*), and more complex environments affect how individuals allocate resources towards cognitive development (*Sheenaja & Thomas, 2011*). In this particular case, however, habitat complexity isn't determined by some environmental cue, but is determined by mate density since females will need to more successfully search for males in a lower density environment, but will need to more accurately distinguish between males in a high density environment. Additionally, individual movement throughout different behavioural trials may be associated to some extent to individual personalities, which may be relevant since juvenile social environments

can affect both how individuals develop and how bold they are (*DiRienzo, Pruitt & Hedrick, 2012*). We thus hypothesized that females reared under different social environment treatments will differentially invest in cognitive ability and that this would manifest in differences in searching performance and the ability to learn.

To test our hypothesis, we reared individual *T. commodus* females in two different social environments: under silence or among recorded heterogeneous male calls. Due to the large amount of support for the effect of environmental heterogeneity on the cognitive abilities in vertebrates and invertebrates alike (*Van Praag, Kempermann & Gage, 2000*), we predicted that individuals experiencing the calling treatment during developmental stages would have increased cognitive abilities compared to those reared in silence. We used two separate experiments to examine whether the juvenile social environment affected individual problem solving, learning, and memory. The first experiment made use of a simple learning task well established for use in neurological and behavioural studies with clear procedural protocols (*Vorhees & Williams, 2006*), the Morris water maze (*Morris, 1981*). It has also been used to successfully demonstrate learning ability in the cricket species *Gryllus bimaculatus* (*Wessnitzer, Mangan & Webb, 2008*). The second experiment examined spatial and route learning in female *T. commodus* in a novel linear maze of original design. In female *T. commodus* specifically, fitness relies on their ability to locate a mate in order to reproduce (*Kasumovic et al., 2011*), therefore the design of the novel maze aims to replicate the process of phonotaxis—directional search for a broadcasting male in order to mate (*Loher & Rence, 1978*).

## MATERIALS AND METHODS

### Rearing treatments

We used female Australian black field crickets (*Teleogryllus commodus*) that were fourth generation lab reared individuals of third generation stock supplemented with 200 individuals collected in March 2014 at Smiths Lake (32°22′S, 152°30′E), NSW. Individuals were reared in communal tubs (75 × 45 × 45 cm) with *ad libitum* food and water until the penultimate juvenile instar. We removed females when they could be sexed (two instars prior to maturity) and housed them in individual plastic containers (5 × 5 × 3 cm) with a single egg carton for shelter, water, and four pieces of cat food replaced weekly. Individuals were checked daily for eclosion to the last juvenile instar after which they were measured (pronotum width), weighed and randomly assigned to one of two rearing treatments according to *Kasumovic et al. (2011)*.

In two separate acoustically isolated rooms, females either experienced the calls of three recorded males, each calling at a different rate, or silence (see *Kasumovic et al., 2011* for greater details on the nature of the recordings used). To create these calls, we used Adobe Audition (version 3.0) to manipulate the inter-call duration (ICD) according to the variation outlined in *Hunt, Brooks & Jennions (2005)*. We recorded calls from three stock males and manipulated two calls from each male to create a calling bout with a mean ICD (0 SD; medium call rate, 18 calls/min) for each male. We then altered the ICD -1 SD of one male to create a high call rate (25 calls/min) and a second male 5 SD to create the low call

rate (12.5 calls/min). In the calling treatment, three speakers (Logitech R-10) were placed on opposite sides of a one metre diameter circle surrounding the females. The speakers played recorded high (25 calls/min), medium (18 calls/min), and low (12.5 calls/min) quality calls at an amplitude of 70dB during the standardised 12 dark hours experienced by all individuals (*Kasumovic et al., 2011*).

To ensure there was no influence of room and speaker location, individuals were moved between treatment rooms daily and the placement of individuals between the speakers was randomised. The silent treatment was isolated acoustically, ensuring the females did not experience any male calls between their final instar and maturity. We checked individuals daily for maturation, and once mature, they were once again measured, weighed and had their development time recorded. Females in Experiment 1 were tested the next day whereas females in Experiment 2 remained in their individual plastic containers but were housed in our communal stock room, where they heard varied adult males calling until testing after day 10, in line with previous studies with the same methodology (*Kasumovic et al., 2011*). All cricket stock, treatment and testing rooms were temperature controlled at 28 °C and maintained on a 12:12 hour light: dark cycle.

## Experiment 1: Morris water maze analog
### *Apparatus*
Derived from the heat maze design introduced by *Foucaud, Burns & Mery (2010)* for use with *Drosophila*, the apparatus consisted of 64 thermoelectric elements (each 4 cm$^2$) aligned in an $8 \times 8$ array (Fig. 1A). This allowed us to create an arena with a floor temperature of $45 \pm 4$ °C, except for a single element maintained at $28 \pm 4$ °C to act as the target zone, similar to previous Morris water maze (MWM) analogs (*Mizunami, Weibrecht & Strausfeld, 1993*; *Mizunami, Weibrecht & Strausfeld, 1998*; *Wessnitzer, Mangan & Webb, 2008*). On top of the thermoelectric elements, we placed a thin copper layer covered with white contact paper (Fig. 1A). This served three purposes: (1) it visually camouflaged the single lower temperature target zone, (2) it ensured an even temperature distribution throughout the arena, and (3) it allowed for easy removal of any possible olfactory cues between trials. We placed a circular wall around the area (30 cm diameter and 30 cm in height) on top of the constructed arena floor to ensure there were no visually identifiable corners and that individuals could not escape the apparatus.

The whole apparatus was covered with a semitransparent white cloth canopy in order to control for unwanted extra-maze cues while allowing for incoming light from a bulb positioned directly above the centre of the arena. Along the bottom edge of the arena wall, corresponding to each axis, four different geometric designs (Fig. 1B) were positioned to act as visual cues that females could use to find the target zone. A web camera (Logitech Pro C920) was positioned at the centre of the canopy so that we could record each trial.

### *Methodology*
On day one or two after maturation, we exposed females to a training session (pretraining trial) where they were placed on the cooled target zone for a 5-minute period under a transparent glass beaker. The diameter of the beaker was large enough to allow the female to explore the area outside the cooled target zone to become familiar with the aversive

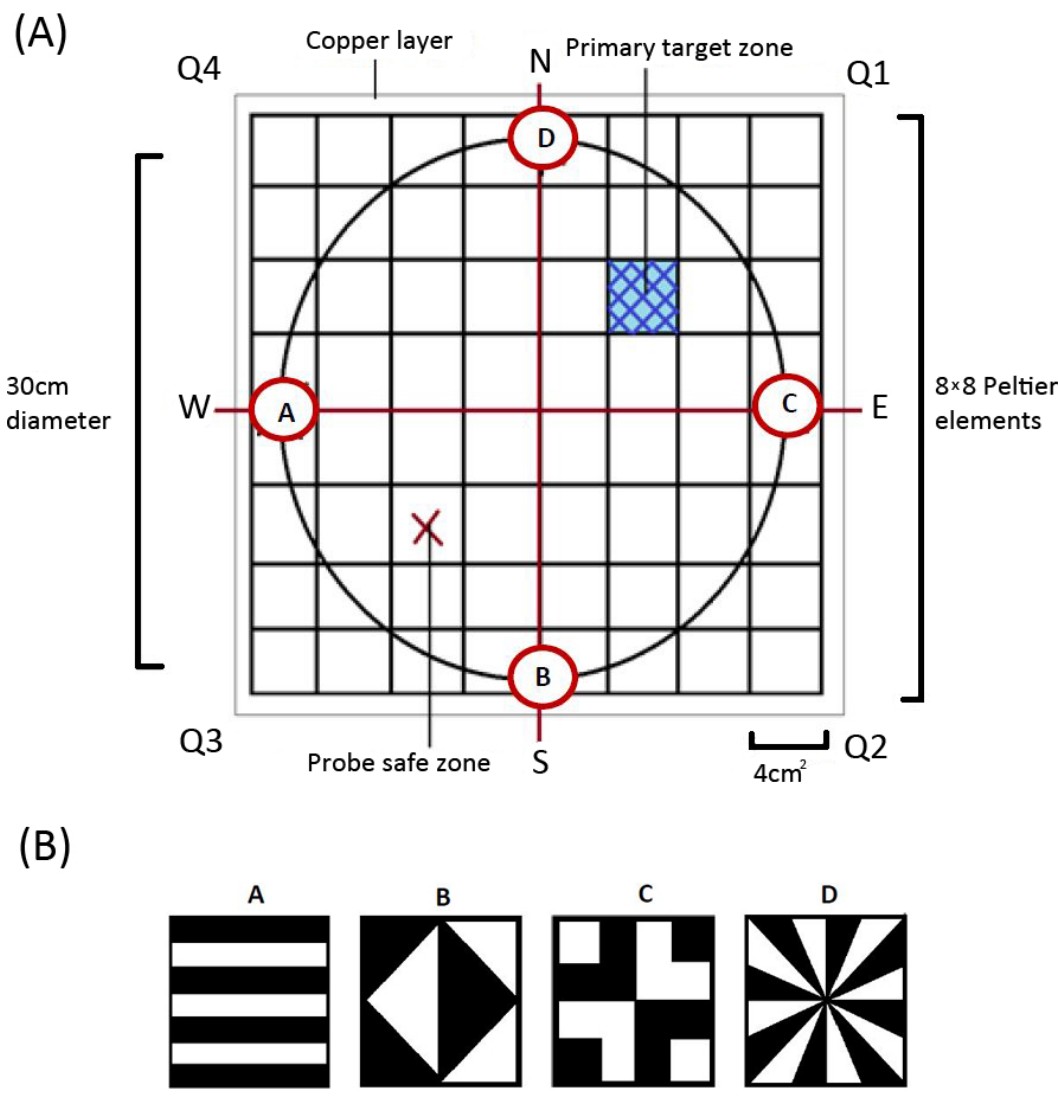

**Figure 1** (A) The design of our heat maze as seen from above. N, E, S, W = The resulting direction of the axes $X$ and $Y$ separating the design into four quadrats (Q1-4) each consisting of $4 \times 4$ Peltier thermoelectric elements. The positioning of each visual cue (A–D) corresponds with each of the 4 axes. (B) The visual cues used in the heat maze positioned along the interior bottom edge of the arena wall.

heated surroundings and orient towards the visual cues. The initiation trial involved removing the beaker and allowing the female to move off the target zone and explore the area in accordance with similar MWM analogs (*Mizunami, Weibrecht & Strausfeld, 1993*). If females did not move off the zone within 30 s, they were gently guided off the target zone with a brush in whichever direction they were facing. These trials were specifically used to train the cricket to the task and were not examined in the statistical analyses.

After the pretraining and initiation trial, we ran six regular training trials where we placed females in front of and facing one of the four visual cues. The specific visual cue females were placed in front of was randomized for each trial to minimise the use of muscle memory or route learning. We then recorded whether females successfully located the

target zone and how long it took females to reach it. The target zone was located in the middle of Q1 for all the pretraining and training trials. We next completed a probe trial which allowed the testing of visual learning. For the probe trials, the target zone was rotated 180° by rotating the moveable arena wall and was thus located in Q3 (referred to as the 'probe safe zone' in Fig. 1). We then completed one last regular training trial on day two to see if any learning was maintained overnight.

Each individual thus received a pretraining trial, a single initiation trial, six regular training trials, a single probe trial, and a single regular training trial on the subsequent day. We completed these trials for a total of 18 individuals from each treatment ($N = 36$). During all trials, the crickets were handled with a transparent glass beaker and transparent perspex sheet to minimise handling stress to the animal. Between all trials of both experiments, individuals were rested for two minutes in a white plastic tub. All trials ended after the female either made contact with and remained on the target zone for a minimum of 30 s or after a maximum of five minutes to minimize the possibility of heat shock. If they did not reach the target zone within five minutes, the individual was guided onto it with the use of a transparent beaker and rested there for 30 s before being removed for the rest period.

## Experiment 2: linear maze
### Apparatus

The novel design (Fig. 2), hereby known as the ball-and-chain linear maze (LM), was derived from the single unit depicted by Ballachey (1934) for use with the white rat. Constructed of white corrugated plastic, each section of the linear maze was 25 cm long and 4 cm high and topped with transparent acetate sheets to prevent escape. This skeletal structure with no attached floor was positioned over replaceable paper towel during trials to remove any olfactory cues. It comprised a single start box leading directly to four identical units, followed by a goal box separated with mesh from a speaker playing a call at a high call rate (25 calls/min) during each trial (Fig. 2). This call was from a unique male that females never heard and it initiated phonotaxis in females. Upon entering each unit, the female encountered a cued wall and two possible pathways, featuring one visual cue (using cue A in Fig. 1B) associated with an open pathway and an uncued wall associated with a blocked pathway.

We recorded the progress through the maze using a web camera (Logitech Pro C920) positioned approximated one metre directly above the centre of the maze. As female *T. commodus* are nocturnal and perform all their searching at night, all trials were performed in a near dark room. Low lighting was provided from a single and consistently lit direction separated from the room by black cloth in order to mimic moonlight. As crickets are unable to see red light, a single red globe was positioned above the maze to facilitate video recordings.

### Methodology

As female *T. commodus* do not reach sexual maturity until seven to 10 days after maturation, we tested a female's performance in the maze 10–15 days following adult eclosion. All individuals were first tested for search behaviour with a single pretraining trial. In the pretraining trial, all pathways of all four units were left open and each pathway was
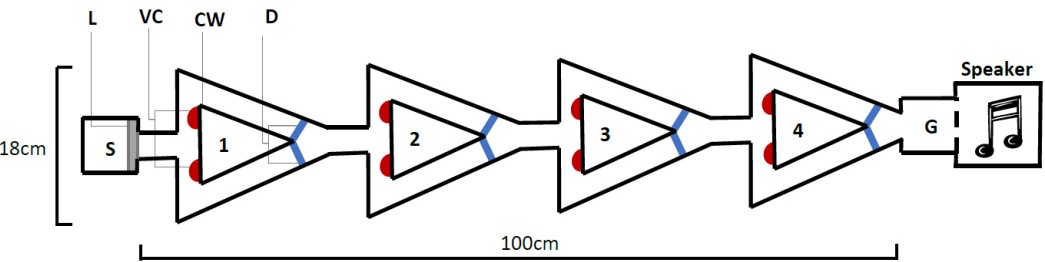

**Figure 2 The linear maze apparatus as seen from above.** Females navigated four identical units (1–4) each 25 cm in length. Individuals began in the start box (S) and the trial began when a female's abdomen crossed the start line (L). They were required to reach the goal box (G), separated from a speaker by a mesh wall. At the beginning of each unit (1–4), females were presented with a single visual cue (VC), placed on the bottom corner of the cued wall (CW) closest to the pathway that gave them access to the correct pathway; the pathway that allowed them to pass out of the unit. Incorrect pathways were uncued and completely blocked with a door (D) on one side of each unit.

associated with a visual cue to allow females to learn that an open door was associated with a visual cue. Twenty females from each treatment demonstrated a level of searching applicable to training. This was defined as whether they approached the mesh wall within a maximum of four minutes during this pretraining trial. We next tested these females for learning with a maximum of seven training trials (depending on their performance), one probe trial and a final test the following day to see if any learning was maintained overnight.

All trials began when the entirety of the females' abdomen crossed the start line between the start box and unit 1 (Fig. 2). If they did not do so within two minutes, they were gently guided out of the box with a brush. All trials ended when an individual either completed the trial by approaching the mesh wall, displayed non search behaviour by remaining stationary for over one minute, or after the maximum time had elapsed. Evidence that an individual had mastered the task and learnt the correct route through the maze was demonstrated by navigating the maze twice in a row with no errors. All training and probe trials had a maximum of five minutes, not including time spent in the start box. For each training trial, one pathway of each unit was blocked off completely by a white corrugated plastic door attached to both walls with white tape. Females were not able to see which pathway allowed them access to the subsequent unit (the correct pathway) or which was completely blocked (the incorrect pathway) because of the design of the maze (Fig. 2). Females would thus need to associate the visual cue with the open pathway. The correct path through the four units was randomly selected for each individual prior to training trials and always contained two correct pathways to the left and two correct pathways to the right in order to minimise the influence of any possible turning preferences. Each probe trial involved the mirroring of the final correct pathway the female was tested with to examine whether they would follow the cues to the mesh wall despite being positioned on the opposite wall. During all trials, the females were handled with a transparent glass beaker and transparent perspex sheet to minimise handling stress. Between all trials of both experiments, individuals were rested for two minutes in a dark, covered white plastic tub.

For each trial, we recorded the length of time it took females to complete the task as well as pauses while searching and the number of times females attempted to dig into a corner or door in an attempt to break through the barriers of the maze. We also counted the number of errors females performed, defined as following the incorrect pathway, following the correct pathway but turning around instead of completing the unit, and backtracking into the previous unit.

## Statistical analysis

All trial video files were renamed and randomised by an external party to ensure a blind analysis. Video recordings of each trial were analysed in real time with the behaviour tracking software program JWatcher V1.0 (a joint venture by Macquarie University and UCLA, 2000–2006).

We calculated a development rate for each individual by taking the inverse of the number of days it took an individual to mature to ensure a normal distribution. We used a MANOVA to examine whether our treatments affected developmental rate, growth, and weight gain. Growth and weight gain were calculated as the difference between measurements taken during the adult and last juvenile instar and controlled for by the final juvenile instar size. We then used individual ANOVAs to explore each of the individual traits.

To examine whether any factors affected success in the heat maze, we first used a linear model to explore whether treatment, development rate, and trial weight predicted the number of successful trials. We only used the six training trials in this examination. We next used a linear mixed model to examine whether treatment, development rate, or trial weight predicted the time to completion for the individuals that successfully completed the trials; we used individual as a random variable in this analysis. We next looked for evidence of learning by exploring whether individuals improved their time to completion across subsequent trials. For this analysis, we used a linear mixed model with treatment, development rate and weight as factors while controlling for trial number and starting position; we also used individual as a random variable. We included individuals that did not complete the trials successfully with the maximum time value possible (300 s). We also used a linear model to examine whether treatment, development rate, and trial weight predicted the time to completion in the probe trial and the training trial on the subsequent day.

We used a generalized linear model (GLM) to examine whether development rate, weight, and treatment affected whether a female successfully completed the linear maze; we used trial as a covariate and added female identity as a random variable. We next used a mixed model to explore whether development rate, weight, or treatment affected the speed at which females successfully completed a trial; we used trial as a covariate and added female identity as a random variable. We used the same type of mixed model analyses to examine whether these variables affected the number of errors, pauses, and shoves during the linear maze trials. We also used a GLM to examine whether treatment, development rate, and trial weight predicted completion success in the mirror trial and the trial on the subsequent day, and a linear model to examine whether treatment, development rate, and trial weight predicted the time to completion of successful trials in the probe trial and the trial on the subsequent day.

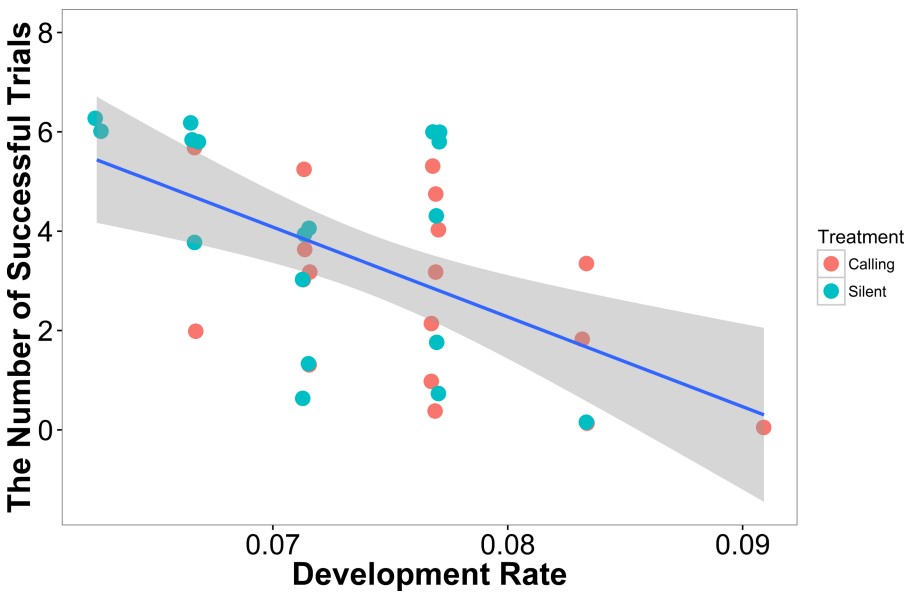

**Figure 3** **The number of successful trials a female completed was predicted by the development rate with females that developed at a higher rate having a lower success rate (N = 36).** Shaded areas are 95% confidence intervals. We minimized the overlapping of points using the "jitter" function in ggplot2.

We used R 3.3.1 to perform all analyses and used the lme4 package (*Bates et al., 2015*) to perform mixed model analyses. We log transformed the number of errors, pauses, and escape attempts (shoves) to improve the distributions. All data and R code can is available on Gihub (https://github.com/latrodektus/Plasticity-and-Learning-R-Analysis).

## RESULTS

### Rearing treatments

We reared a total of 76 females in the two rearing treatments. There was no overall effect of rearing treatment on development (MANOVA: $F_{3,72} = 0.37$, $P = 0.77$). Subsequent ANOVAs for each trait also demonstrate no effect of treatment on development rate ($F_{1,74} = 0.83$, $P = 0.37$), growth ($F_{1,74} = 0.12$, $P = 0.73$), or weight gain ($F_{1,74} = 0.07$, $P = 0.79$).

### Experiment 1: Morris water maze analog

In our linear model, the number of successful trials a female completed was predicted by development rate ($t = -3.47$, $P = 0.001$), with females that developed more quickly having a higher failure rate (Fig. 3). Neither weight ($t = 1.02$, $P = 0.31$) nor treatment ($t = 1.02$, $P = 0.31$) predicted a female's success rate.

In our linear mixed model, we found no evidence that treatment ($t = -0.05$, $P = 0.96$), development rate ($t = 0.09$, $P = 0.93$), or weight ($t = 1.70$, $P = 0.10$) predicted the completion time of the successful trials. We also found no evidence that individuals improved in how quickly they completed the heat maze over subsequent trials (Table 1, Fig. 4), suggesting that individuals did not learn during the trials. Development rate was nearly significant ($F = 3.48$, $P = 0.07$) likely because the addition of individuals that did

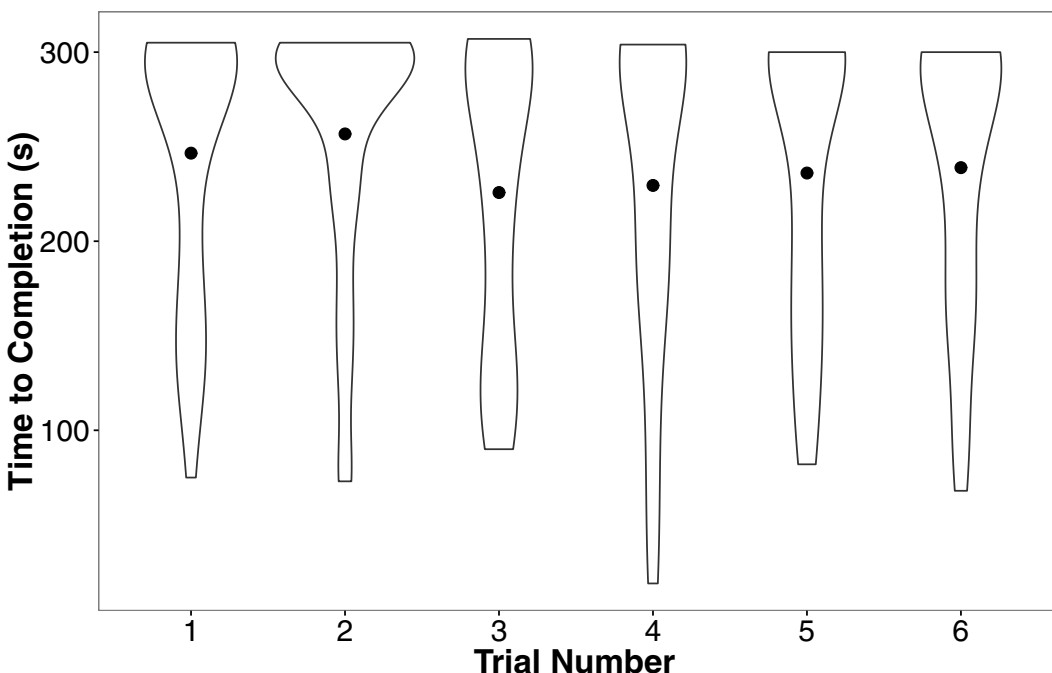

**Figure 4** **Violin plots of the completion time for individuals in the six regular training trials.** The width of the plot signifies the probability density of the data at different values and the dot is the mean. There is no evidence for improvement over the trials. $N = 36$ in each trial.

**Table 1** **The results of the linear mixed model examining whether time to completion varied across trials as a consequence of the trial, phenotypic traits, and the starting position.** Since development rate was an important predictor of success, the interaction term specifically explores whether individuals with different development rates varied in their learning.

|  | DF | F | P |
|---|---|---|---|
| Treatment | 1, 31.22 | 1.36 | 0.25 |
| Development rate | 2, 30.77 | 3.48 | 0.07 |
| Weight | 2, 30.47 | 0.52 | 0.47 |
| Trial | 5, 156.94 | 1.42 | 0.22 |
| Start Position | 3, 160.41 | 1.01 | 0.39 |
| Development rate × Trial | 5, 156.56 | 0.37 | 0.87 |

not complete the trials developed significantly faster (Fig. 3). There were no factors that predicted female performance in the probe trial (All $P > 0.21$) or the training trial on the subsequent day (All $P > 0.18$).

## Experiment 2: linear maze

Females from both treatments successfully completed a similar number of linear maze trials (silent: $6.55 \pm 0.20$, calling: $6.65 \pm 0.22$; $N = 40$). Treatment, development rate, or weight did not predict whether females completed the linear maze successfully (All $P > 0.50$). Of the females that successfully completed the linear maze trials, there was no effect of

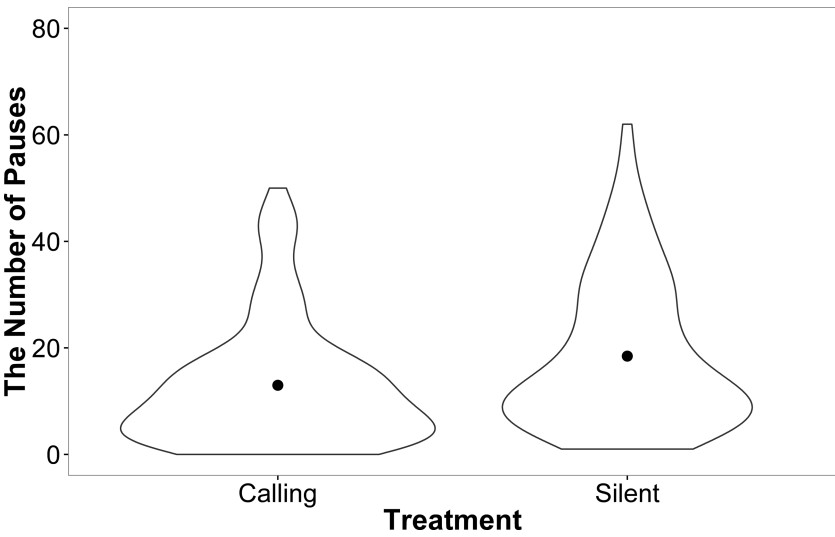

**Figure 5** A violin plot demonstrating that a small number of females within the silent treatment pause more often than females from the calling treatment during linear maze trials. The width of the plot signifies the probability density of the data at different values and the dot is the mean.

treatment, development rate, or weight on the speed with which females completed the trial (All $P > 0.20$).

The treatment, development rate or weight did not affect the number of errors that females made during the trial (All $P > 0.45$) or the number of attempts to escape the trial (All $P > 0.30$).

Female motivation and search strategy were explored by examining the number of pauses (remained stationary for a count of 3 s) made and amount of shoves events towards walls and doors during each trial. There was no effect of development rate ($F = 0.00$, $P = 0.99$), weight ($F = 0.14$, $P = 0.71$), or trial ($F = 0.50$, $P = 0.81$) on the number of times females paused while searching in the linear maze. However, females from the silent rearing treatment paused more often than females from the calling treatment (silent: $2.71 \pm 0.07$, calling: $2.28 \pm 0.09$; $F = 9.86$, $P = 0.004$). This effect, however, seems to be driven by a small number of individuals from the silent treatment that paused more frequently (Fig. 5).

There was no effect of development rate, weight or treatment on whether females successfully completed the mirror trial (All $P > 0.11$) or the trial on the following day (All $P > 0.11$), nor the speed that successful females completed the probe trial (All $P > 0.30$) or the trial on the following day (All $P > 0.21$).

## DISCUSSION

Phenotypic plasticity has been shown to be present in almost all organisms (*West-Eberhard, 2003*). Investment into cognitive development and ongoing maintenance are particularly plastic traits as they are energetically costly (*Dukas, 2013*). Although cognitive phenotypic plasticity is largely explored in vertebrate subjects (*Van Praag, Kempermann & Gage, 2000*) and in response to environmental factors (*Kotrschal & Taborsky, 2010*; *Sheenaja & Thomas,*

*2011*; *Girvan & Braithwaite, 2000*), the effect of immature experience on invertebrates and the role this has on cognitive development is unclear (*Durisko & Dukas, 2013*). The aim of this study was thus to examine the effect of the immature social environment on the cognitive performance of the Australian black field cricket (*Teleogryllus commodus*). We used two different learning paradigms to examine cognitive ability, including learning, memory and problem solving between individuals reared in heterogeneous calling and silent acoustic treatments. There was no evidence of learning regardless of treatment or experiment.

## Experiment 1: Morris water maze (MWM) analog

Despite this lack of treatment effect, there was evidence of differences in cognitive performance between individuals when using the Morris water maze analog. Individuals that developed at a slower rate showed improved performance, as measured as the number of successful trials, indicating that juvenile development rate may have a greater role in determining cognitive ability than the social environment experienced while developing. This suggests that there are possible cognitive advantages in taking longer to mature, suggesting individuals are investing more resources into cognition during this time or that such costly resources take longer to develop. Studies on certain bird species show that those that thrive in difficult, increasingly urbanised areas, are those with longer periods of incubation and fledging (*Pocock, 2011*; *Maklakov et al., 2011*), suggesting that increased development time might be necessary for increased cognitive development.

The use of the MWM analog to test visual and spatial learning ability in invertebrates (e.g., *Periplaneta Americana*; *Mizunami, Weibrecht & Strausfeld, 1998*, *Drosophila melanogaster*; *Foucaud, Burns & Mery, 2010*) such as crickets (*Gryllus bimaculatus*; *Wessnitzer, Mangan & Webb, 2008*) has shown previous success. Despite this, during this study, female *T. commodus* did not show any improvement across trials during the MWM task and did not learn the location of the target zone. As a consequence, it is not surprising that there was no difference between the training trials and the probe trial, as there was no evidence that individuals used the visual cues to locate the target zone. Although it is possible that the absence of learning may be due to the lack of ecological relevance of this paradigm to the fitness of female *T. commodus*, the fact that it was previously successfully used in crickets (*Gryllus bimaculatus*; *Wessnitzer, Mangan & Webb, 2008*) and cockroaches (*Periplaneta Americana*; *Mizunami, Weibrecht & Strausfeld, 1998*) suggests that other unknown factors are attributable. Previous studies did not supply information concerning the age or rearing protocols used for their individuals so it is difficult to make a true comparison. The only known variation was the apparatus used to control the temperate of the aversive surface. Rather than using varied water flow, this experiment utilised thermoelectric Peltier elements and was modelled off that of *Foucaud, Burns & Mery (2010)* for use in *Drosophila*. Similar to this study, despite the long success of *Drosophila* and other invertebrates in modelling learning processes in a sexual behaviour context (*Dukas, 2006*), multiple attempts at altering cognitive development through early life variability has also shown mixed results (*Durisko & Dukas, 2013*).

## Experiment 2: linear maze

There was no evidence that female *T. commodus* learn within the novel linear maze apparatus used during Experiment 2 or that females differed in their searching success as a function of development rate. After maturation, all individuals used in Experiment 2 were housed in a communal stock room along with a variety of calling males. Individuals were housed communally as pilot studies demonstrated that if individuals were housed in isolation following maturation in order to fully preserve treatment effects, they showed considerably less search behaviour when tested. However, this communal rearing may have overridden our treatment effect on cognitive ability for the individuals used during Experiment 2.

The only behavioural trait that demonstrated variance between the two treatments is a difference in the number of pauses with individuals from the calling treatment pausing less on average. This result, however, needs to be taken with caution as it was likely driven by a small subset of females that paused for longer periods of time. Anecdotally, *T.commodus* exposed to this specific calling treatment produce more eggs during their lifetime than those in the silent treatment (*Kasumovic et al., 2011*). It would be interesting to explore whether a female's searching strategy and intensity is affected by their reproductive capacity in future studies.

Despite the novel nature of our linear maze paradigm and the difficulties in design that have been discussed, we included this experiment in the final study to highlight the difficulties in creating a biologically relevant learning paradigm (*Timberlake, 2002*). Many learning studies aim to control the motivation and consistent performance of the animal models by depriving them of food and water and using these necessities as rewards for the successful completion of a task (e.g., *Matsumoto & Mizunami, 2000*). However, these nutritional deprivations have their own influences on cognitive performance (*Rowe & Healy, 2014*), which is why such motivations were avoided for this study. Although rewarding the completion of a learning task is a typical aspect of many learning studies and has been shown to increase learning and memory retention (*Nakatani et al., 2009*), it was not applicable to the design used in our study. Thus, a possible improvement for further study would be to incorporate a reward system. For this study, although females located the potential male, females were never allowed to mate with this 'phantom' male, as mating would have affected further performance and any learning effects. This again highlights the difficulties in designing a learning paradigm that ensures consistent completion as well as application to true biological behaviours (*Rowe & Healy, 2014*).

## Treatment effects

In previous studies with *T. commodus*, the quality and density of calls that males and females heard prior to maturity affected how individuals invested towards different traits (*Kasumovic et al., 2011*; *Kasumovic, Hall & Brooks, 2012*). This plasticity, however, was not seen in this study despite these same treatments having been shown to influence development rate (*Kasumovic et al., 2011*; *Kasumovic, Hall & Brooks, 2012*). It is unclear as to why this study found no difference in developmental trajectories, though possibilities include differences in overall stock population density and low sample sizes compared with

the previous study (*Kasumovic, Chen & Wilkins, 2016*) as well as other abiotic factors such as seasonal differences that may have an unknown impact on stock development. Although variation in stock density and seasonal variation were not noted as issues in previous replications (e.g., *Kasumovic et al., 2011*; *Kasumovic, Hall & Brooks, 2012*), individuals in previous studies all experienced some recorded calling during developmental stages, though at varied qualities and densities. It may be that unknown factors such as seasonal variation and tactile density in communal rearing tubs may be more important for cueing developmental trajectories in the absence of available calls (i.e., in silence). Future studies are needed to explore this possibility and what other factors may be acting on these developmental trajectories. Additionally, as cognition is such an energetically costly resource, its relevance may need to be cued to each individual for a longer time than needed for other altered developmental trajectories. Therefore, a possible future study might involve exposure to treatments at an earlier lifecycle stage in order to prompt a treatment related change in cognitive development. It is also possible that cognitive development is not a trait that can be directly influenced plastically by environmental cues.

It is suggested that learning paradigms not fine-tuned to reflect the biology of the animal model are not an accurate assessment of learning ability in the wild (*Timberlake, 2002*). Although crickets have shown learning ability in other areas, includes reward recognition (*Matsumoto & Mizunami, 2000*), lifetime olfactory memory formation (*Matsumoto & Mizunami, 2002*), visual context dependent odour association (*Matsumoto & Mizunami, 2004*) and paired odour learning (*Matsumoto & Mizunami, 2006*), these experiences likely have less of a fitness association in nature compared to mate location (*Rowe & Healy, 2014*). In order to properly explore the effect of early life social environment on the investment into learning and memory in *T.commodus*, an alternate learning apparatus would be needed. Individual variation would need to be minimised through increasing motivation, possibly through mate choice learning or with the use of novel calls for each trial to maintain interest of the female. An examination into the differences in learning ability across varied ages would also be beneficial in the examination of learning and fitness due to possible differences between life stages. However, this may prove difficult as varied life stages would require differing ecologically relevant learning tasks which would be difficult to compare. Investigation into wild population cognition is a growing field that, despite challenges, would be enormously beneficial to the understanding of how cognitive abilities may have evolved and contribute to ongoing evolutionary processes (*Roth, Krochmal & Németh, 2015*).

## ACKNOWLEDGEMENTS

We would like to thank Heather Try for all her help in cricket rearing, and the reviewers for all their helpful comments.

### Funding

This work was supported by an ARC Future Fellowship to Michael Kasumovic (ARC FT140100115). The funders had no role in study design, data collection and analysis, decision to publish, or preparation of the manuscript.

### Grant Disclosures

The following grant information was disclosed by the authors:
ARC Future Fellowship: ARC FT140100115.

### Competing Interests

The authors declare there are no competing interests.

### Author Contributions

- Caitlin L. Anderson conceived and designed the experiments, performed the experiments, analyzed the data, wrote the paper, reviewed drafts of the paper.
- Michael M. Kasumovic conceived and designed the experiments, analyzed the data, contributed reagents/materials/analysis tools, wrote the paper, prepared figures and/or tables, reviewed drafts of the paper.

### Data Availability

Raw data and code are available on Github: https://github.com/latrodektus/Plasticity-and-Learning-R-Analysis.

### Supplemental Information

Supplemental information for this article can be found online at http://dx.doi.org/10.7717/peerj.3563#supplemental-information.

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
