# Peer review of "Development rate rather than social environment influences cognitive performance in Australian black field crickets, Teleogryllus commodus"

_PeerJ, doi:10.7717/peerj.3563_

## Round 0.1 · original submission · Major Revisions

· Academic Editor

Major Revisions

Please revise your manuscript paying close attention to the criticisms of the reviewers. I am especially concerned about the age of the tested crickets and their ability to hear songs, so make sure to address this carefully. Please be advised that your revised manuscript will be sent back to the reviewers, and will only be accepted if both they and I feel you have addressed all of the criticisms successfully.

Reviewer 1 ·

Basic reporting

The Introduction should be revised to better incorporate the natural history of the study organism, as well as the rationale, and predictions for the study. There is a fair amount of extraneous information and the introduction is somewhat disorganized. Some specific suggestions follow:

L42-45: It is not clear to me the relevance of this sentence to the author’s study. Suggest replacing it with something suggesting the general direction of the study.

L46-55: This could all be simplified and combined with L56-79. The authors might consider discussing how use of environmental cues (e.g., food, mate cues) as a juvenile influences adult cognitive abilities and how this is seemingly a fitness benefit. There is no need to go into so much detail on various environmental situations not relevant to the study, nor to review what cognitive ecology now focuses on. These paragraphs instead should bear on the usage of the study organism and its ecology.

L80-L88: There are several problems with this paragraph. One: the authors make a distinction between phenotypic plasticity and learning without explaining how they are different. Behaviour is part of the phenotype of the organism. Thus learning, which enables plasticity in the behavior of the organism, is an example of phenotypic plasticity, not separate. L83-L88 is likewise poorly explained (e.g., what physical attributes are we supposed to care about?) and does not give any hints as to why the particular study organism is ideal.

L89-96: all of this information should have been integrated into prior paragraphs, first in a more general setting, and then in a more specific. I.e., why the study organism is typical and thus fit for experimentation here.

L102-106: The authors should briefly discuss how these experiments are pertinent to the ecology of the organism. The experiments seem to come out of the blue, just because they have been used with other organisms.

Several of the figures lack sufficient detail:

Figure 3: it is not possible to have half a trial (or other non-integers). Please adjust the Y axis of the graph to include only integers. Likewise I could not find anywhere in the text the units for development rate, nor how it was determined (which should be included in the Methods).

Figure 3: 19 data points are included, but only 18 individuals were reported to be tested. What is going on here?

Figure 4: time to completion in what? Presumably seconds. Likewise, what are the sample sizes for each trial?

Figure 4: if there are 6 regular training trials, why are trials labeled 2-7?

Figure 5: does the figure report the mean number of pauses for each animal, or the total across all animals and treatments? If the latter, that would involve pseudoreplication, since animals that completed more trials would presumably contribute more pauses. Please clarify in the text.

Minor points:

The language used throughout the manuscript is generally clear and understandable. However, occasionally the English requires revising either because of grammatical issues or to improve clarity. Please consider making a careful sweep of the text. Some examples include:

L32: Cognition encompasses learning and memory, so best to adjust sentence accordingly: “Learning and memory in particular are drivers of…”

L35: rapid change in what?

L35: rapid change in what?

L41: and/or?

L43: “and is more apparent”; what is more apparent? The trade-offs?

L46: change “individuals within a species” to “conspecifics”

L48: “made necessary”; avoid agency or intentionality.

L56: If it is more recent work, then the references provided in this and the prior paragraph do not bear that out.

L59: change “pre-emptively assess” to “predict”

L112: replace “second last” with “penultimate” here and elsewhere

L274: improved what over time?

L275: is a comma missing between "treatment" and "development rate"?

L316: "nearly significant" is a misleading term – especially given the use of multiple comparisons . Would recommend to simply state ****

L322: Correct to "The results of the linear..."

L323: Correct to "varied across trials as a consequence..."

L329: "despite these same treatments having been shown"

L333: successfully completed?

L358: what is especially true? This sentence does not follow from the first sentence, which seems also to be a truism.

L385: "e.g."

L395: what is meant by success? Showing evidence of learning? Please clarify.

L381-382: Why should seasonal differences account for these differences?

And similar issues.

Experimental design

The research question is meaningful, but as discussed in more detail earlier, could use a more on point Introduction . I.e., integrating the natural history of the organism more carefully to explain the relevance of the particular experiments used and the validity of the questions being asked in relation to the biology of the organism.

There appear to be several potential major flaws with Experiment 2 that makes its inclusion in the manuscript seemingly problematic:

1. Are females in the dark able to use the provided visual cues? No citation is provided. It seems also no control experiment was performed in the absence of visual cues to determine whether the crickets could navigate in their absence.

2. Additionally, the open/closed door may have provided a cue that could have been used, as the authors had no protocol for removing crickets that investigated both pathways at each stage (which could potentially have allowed crickets to assess both paths without learning the cue). Although the authors tracked errors, this did not appear to be one of the errors being tracked. The authors do say that females were not able to see which pathway led into the box, but if one pathway was blocked off, they should be able to see that I would think... Hopefully this is a matter of unclear writing.

3. Why weren't individuals used in Experiment 2 housed with non-calling juveniles if, as the authors say, housing communally is very important to the development of search behavior? This would presumably have preserved treatment effects while avoiding issues associated with housing in isolation following maturation. This seems to be a fatal flaw in the design of Experiment 2, because by housing with calling individuals, effects of the juvenile treatment may have been obviated (as the authors themselves note within the Discussion).

In several places key methods are also not described nor justified:

Speed to complete a trial is confounded by movement speed of the organism, which is not necessarily a cognitive factor. Could the authors analyze path length or a similar category, which would be more associated with spatial learning?

Nowhere is development rate defined or its methodology described.

L138: An ingenious contraption! However, nowhere is this floor temperature justified. Is this hot enough to motivate the organisms to seek shelter? Is there a citation to show this? If this temperature is not aversive, perhaps some crickets have different thresholds at which they are motivated, which could account for some not cooperating.

L176: Perhaps critically, did the time spent on the target zone differ significantly from time spent still in other regions? If it didn’t, completion of a trial might not be a guided behavior.

L199: So was the paper towel replaced after each trial then?

L255: How was a "pause" defined unambiguously? If this is being used in the data analysis, this must absolutely be defined clearly and without subjectivity.

Nowhere are the visual cues that are used in experiment 1 justified. Are these sufficiently large/distinct for the organisms to discriminate? Without such justification, a reader might come to the simple conclusion that the reason results of learning assays were negative were that the cues used were not discriminable.

L294: "to improve the distributions" - do the authors mean that log transformations normalized residuals? Otherwise, what is meant by improving distributions? Transformations should not be performed arbitrarily.

Minor points:

L221; Provide a citation for crickets being unable to see in red light?

L163: Please call it just a “pretraining trial”. The authors use “training trial” later, which makes its earlier use misleading and unnecessary.

L213: Instead of “uncued”; “presented no cue”

L294: How was an "escape attempt" defined unambiguously? Please also refrain from redefining the term as “shoves”; escape attempt is clear, but “shove” is not.

Sample sizes are only listed once, with the second experiment in fact only listing the total sample size and no sample size per treatment; the reader is left to assume that an equal number of individuals was tested in each treatment. Please repeat sample sizes at least within the figures and within the methods (for the first experiment this is done clearly, for the second it is not).

Validity of the findings

The statistics appear rigorous and appropriate with a single exception: it appears the authors are making multiple comparisons on the same dataset, yet no Bonferroni correction and associated alpha value is described. This becomes important only in the case of the result that appears to trend to significance (which the authors lay importance on).

The Discussion is very thorough and clearly discusses most of the methodological shortcomings and is clearly linked to the research. However, it still could use some revising, shortening, and clarification. In particular, the Discussion jumps around considerably across paragraphs. It thus does not appear to follow a particular flow and is difficult to navigate. For example, L393-443 first addresses potential issues with the MWM analog, then jumps to issues and disparities between the two experiments, then issues related to motivation and use of classical conditioning in both experiments. Further more specific issues are listed below:

L367-370: the argument about juvenile development rate having a greater role than social environment seems highly speculative, as the authors cannot compare the two experiments due to testing the crickets at different ages as well as radically different conditions just before testing occurred.

L370-371: or that developing the sensory/memory equipment requires more time. There is some evidence across taxa that individuals with slower development have better cognitive capabilities (and/or bigger brains): e.g., [Iwaniuk, A.N, Nelson, J.E. 2003. Developmental differences are correlated with relative brain size in birds: a comparative analysis; Devlin, R.H., Vandersteen, W.E., Uh, M., and Stevens, E.D. 2012. Genetically modified growth affects allometry of
eye and brain in salmonids]

L390-392: On the contrary, it does not seem that the authors have been able to find a clear pattern related to all aspects of cognition that this study was intended to cover, except with respect to the more minor (but statistically significant) result that crickets that completed more trials were those that had developed more slowly. Absence of evidence is presumably not evidence of absence. This sentence on the whole also seems unclear.

L400: If an argument about the ecological relevance of this paradigm is going to be made, it needs to be justified. Do these crickets experience such heat stressed environments? Do they move about during the day? Some natural history discussion (which is nearly entirely absent) would come in handy to ground this work.

L428-429: It is not clear to me what is meant by this sentence in relation to the prior sentences. At first it seems that the authors are arguing that differences in motivation could be due to the treatment. Then it becomes about egg production? Where does egg production come into play? There is something missing here.

L426-L436: The discussion about motivation seems speculative. The number of pauses (which, as a term, is not defined…) could be due to sensory integration differences, for instance (e.g., the sensory thresholds of the crickets in the quiet treatment to calls could be different than those in the calling treatment). Unless the authors can state why pauses indicate motivation definitively and with citations, I would suggest to remove these arguments. Furthermore, the authors in the results state that the result is a function of a few females, so this result’s generality to the treatment as a whole is rather undercut.

L432-443: the authors make a well-reasoned argument for how incorporating a reward system (e.g., operant conditioning) would be valuable. However, given that many other studies across taxa have used MWM and other non-operant tests to successfully show learning, the argument that MWM analog might have showed weak learning because it was non-operant seems somewhat unconvincing. Presumably crickets, like other animals, have to learn even in the absence of a food or mate reward; some integration of such information into the text could be valuable.

Minor points:

One potentially obvious additional discussion point could be whether exposure to the treatment during only the penultimate instar was sufficient to generate a change.

L372-375: These sentences do not seem quite on point. The developmental patterns in this study were apparently unrelated to variation in rearing environment.

L381: Why would seasonal differences affect the results? Could the authors point to any citations that would support such a supposition?

L393: The authors may wish to repeat what the MWM acronym refers to, as it has been many pages since the acronym was first provided in the methods.

Comments for the author

I commend the authors for taking the time and effort to submit work that includes null results. The authors clearly spent significant time crafting their arguments. The manuscript is generally quite digestible.

The main result of this study appears to be that development rate, but not rearing environment, affects cognitive performance (as assessed by number of successfully completed trials); an interesting result. However, a clear objective explanation for how a trial was completed successfully needs to be provided: the authors use time spent in the target zone, but do not describe whether time spent anywhere else in one place was fundamentally different.

Aside from this result, the second of the experiments apparently has significant methodological flaws that could have accounted for some or all of that experiment’s null results. I am not entirely convinced that this second experiment belongs in this manuscript because of these issues; reporting of null results is very desirable, but if the methodology has significant flaws it is difficult in my opinion to justify including those null results. Furthermore, a comparison between the first and second experiments' results is probably not possible.

Other aspects of the methodology will also need to be explained more clearly. Without further clarification from the authors it is difficult to be sure even in the case of significant results (e.g., on trial success and pause frequency) whether data was collected entirely objectively.

Reviewer 2 ·

Basic reporting

The text is clear, the hypothesis and context nicely described. The results are straightforward and well presented.

Experimental design

The question at the origin of this study is of high interest and the chosen experimental procedure relevant to solve the question. The authors did not find the interaction between early social environment and cognition they were looking for but reported honestly both the possible technical or experimental explanations.

Validity of the findings

My main concern with this manuscript is that it seems both in the introduction and in the discussion that the authors were excepting a given interaction between the cognitive abilities and the social treatment and therefore interpret the negative result as a potential experimental failure without really consider that this interaction may simply not exist for such cognitive tasks. Of course, the authors may be right but they should develop more the alternative hypothesis in the discussion.
I was also surprised by the way the influence on cognition of environmental conditions at the early stage is presented at multiple occasions in the introduction (e.g. l 58 to 60 or l.66-67): this influence is presented as an active 'choice' of the animal to invest or not in developing its cognitive skills. Even if I suppose the authors did not really mean that but rather that such influence is adaptive given the cost of cognitive functions, I am not sure I am sharing this view. From the neurobiological literature, it seems to me that environment enrichment stimulate brain connectivity due to enhanced perceptual or social interactions thus maintaining much more synapses and brain faculties that would be destroyed if not used (pruning effect). I suggest to the authors to also present this alternative explanation with some references. Both explanations may be true either at different stages or different environmental conditions or animal species but I did not see any argument to favor one over the other in your text.

Reviewer 3 ·

Basic reporting

The basic reporting requirements for this paper are mostly met. However, I did not see mention of where any raw data would be deposited and it was not available with the text. Please make it available to meet the standards of the journal.

In Figure 1, there is a block labelled as the “probe safe zone,” but I could not find an explanation of what it is in the text or the figure legend. Please clarify it’s function.

Experimental design

I have a fairly major concern regarding the treatment groups used in Experiment 1. If I am understanding it correctly, the authors trained and tested female crickets in the heat maze 1-2 days after eclosion. This testing was following different acoustic environments during their last 2 nymph stages. My concern is as to whether the nymphs were capable of “hearing” a 70dB playback. I know that it has been standard protocol to isolate crickets into these acoustic treatments in pre-adult stages, but Popov et al 1994 J. Comp Physiol A 175: 165-170 tested last instar G. bimaculatus cuticular vibration in tissue just above the future posterior tympanum and found no difference as compared to regular cuticle. The auditory sensilla are present in juveniles, but the posterior tympanum is absent, so juvenile sensitivity to conspecific adult sound frequencies is actually very poor (see Ball and Hill 1978) . While it is possible that T. commodus is detecting some auditory stimulation during the juvenile incubation, it is also likely that most of the treatment effect in all of these studies comes from the adult treatment experience. In the studies cited where a response to this acoustic manipulation was found, adults were tested after having spent at least 10 days in their treatments. This could explain why there was no effect of acoustic environment – essentially that there was no treatment imposed on these crickets. Please address this issue in the manuscript, particularly with respect to your interpretation in the discussion. I would also appreciate a justification included in the methods for deciding to use females at such a young age (when they are typically not even reproductively mature yet) for the assay.

The methods with respect to rearing of the females used in Experiment 2 are somewhat unclear. Were all females (regardless of acoustic rearing treatment group) being housed in rooms with calling males (lines 419-421)? Were the experimental females being housed in group containers? If so, how many individuals were housed together? How large were the containers? Isn’t it at least worth considering that social interactions would affect the response to the acoustic treatment? And rate of development/growth?

I am also a bit confused as to how you calculated developmental rate. Please explain this in both the text and in the legend of Figure 3. Why not simply use the # of days between reaching the last juvenile instar and the final eclosion?

Validity of the findings

It is difficult to interpret the validity of the findings and their interpretation until the concerns above are addressed. Specifically, how were the crickets in Experiment 2 housed? Were the crickets in Experiment 1 actually exposed to different treatments, or were they incapable of detecting the song being played?

---

## Round 0.2 · Minor Revisions

· Academic Editor

Minor Revisions

Thanks so much for your careful attention to detail in the revisions. This is an excellent manuscript. I have just a few requests for further (minor) revisions, as follows:

1) you may want to cite DiRienzo et al:
DiRienzo, N., Pruitt, J., & A.V. Hedrick. Juvenile exposure to acoustic signals from conspecifics alters developmental trajectory and adult personality. Animal Behaviour 84: 861-868. I think it is relevant.
2) Please describe the High, Low and Medium quality songs used.
3) Please describe the song used in the linear maze experiment.
I think you can make these changes easily and they will not significantly impact the swiftness of publication. Thanks.

#S taff Note: It is PeerJ's policy that any suggestions to cite references (e.g. point 1) are optional. Only cite articles if you agree the citations are relevant to your work. #

---

## Round 0.3 · accepted · Accept

· Academic Editor

Accept

Hi, thanks so much for the further revisions. Congratulations on a great paper.